# Impacts of Risk Allocation on Contractors' Opportunistic Behavior: The Moderating Effect of Trust and Control

**Yilin Yin [1], Qing Lin [1,\*], Wanyi Xiao [1] and Hang Yin [2]**

[1]  School of Management, Tianjin University of Technology, Tianjin 300384, China; yinyilin575@163.com (Y.Y.); zhjmd@sina.com (W.X.)

[2]  Department of Construction Management, College of Management and Economics, Tianjin University, Tianjin 300072, China; yinhang@tju.edu.cn

\*  Correspondence: 173103402@stud.tjut.edu.cn; Tel.: +86-180-2235-8100

**Abstract:** In construction projects, contractors often exhibit opportunistic behaviors, which harms the project performance, and risk allocation between clients and contractors affects the contractors' opportunistic behaviors (strong and weak). In this study, a structural equation model was built to explore the impacts of risk allocation on opportunistic behavior and the moderating role of trust and control through an empirical test using a recovery questionnaire with 342 interviewees. The results show that the greater the risk contractors take, the stronger their opportunistic behavior is. Trust has a significant inhibitory effect on both strong and weak opportunistic behaviors caused by risk allocation, while control has a significant inhibitory effect only on strong opportunistic behavior caused by risk allocation. This study enriches the research on the governance mechanism and construction management of opportunistic behaviors and provides management suggestions for risk allocation and control measures of such behaviors.

**Keywords:** risk allocation; opportunistic behavior; trust; control

## 1. Introduction

In construction projects, clients often face loss caused by contractors' opportunistic behavior, such as cutting corners and causing delays [1]. In order to reduce the disputes between clients and contractors during the execution of construction projects and restrain contractors' opportunistic behavior, contracts will clearly stipulate the rights and obligations of the parties and allocate the risks reasonably [2]. However in practice, by controlling prices when designing contracts, clients usually assign most of the risks to contractors [3]. Shi et al. [4] believed that unreasonable risk allocation would increase the incentive for contractors to adopt opportunistic behaviors. In order to obtain or enhance the capacity to prevent the occurrence of unforeseen risk in the future, contractors will exhibit opportunistic behaviors to obtain additional profits in advance to prevent future risks. Some scholars also believe that such unbalanced risk allocation may lead contractors to cut corners or make changes and claims at a later stage to obtain profits [5]. To restrain the negative relationship between unreasonable risk allocation and contractors' opportunistic behavior, studies have focused on how to restrain opportunism through reasonable risk allocation. For example, Jin [6] established a neural fuzzy model for effective risk allocation, Zhang [7] introduced the swing option approach to hedge risks, and Asheem and Shrestha [8] studied how to modify unreasonable risk allocation by investigating water projects in China. Su [9] and Jin [10] provided the factors that need to be considered in reasonable risk allocation. Obviously, few scholars have revealed the influence mechanism between risk allocation and opportunistic behavior.

To solve this problem, this paper takes project governance theory as the theoretical basis. The existing governance mechanisms can be defined as two types: one based on transaction cost economics (TCE), proposed by explicit and formal governance mechanisms, and one based on the theory of implicit, informal governance mechanisms. The former is represented by contracts, and the latter by specified relationships [2,11]. Therefore, when contractors take too many risks to produce an opportunistic tendency, effective control of contracts can raise the cost of their opportunism so as to dilute the relationship between unreasonable risk allocation and opportunistic behaviors. Soft factors such as trust can also prompt contractors to pursue long-term benefits and inhibit this relationship [12].

This paper aims to evaluate the impact of risk allocation on the opportunistic behavior of contractors and discuss the moderating role of trust and control by using a structural equation model. The research results can provide clients with guidance on how to reduce opportunistic behaviors of contractors. Clients can assign appropriate risks to contractors based on the present research, and control project implementation by means of management in order to achieve smooth implementation of projects.

The remainder of this paper is organized as follows. Section 2 introduces the literature review. Section 3 presents the conceptual framework and hypothesis development. Section 4 presents the methodology. Section 5 reports the results and discussion. The paper ends with implications for theory and practice as well as concluding remarks.

## 2. Literature Review

### 2.1. Risk Allocation

Risk allocation refers to the division of responsibility for specific risks in various hypothetical situations [13]. In a construction project, risk allocation refers to the process of assigning risks to both parties of the transaction, clarifying that specific risks shall be borne by one or both parties [14]. Most existing studies focus on the design of risk allocation terms of contractors and empirical research on risk allocation schemes of completed projects based on case analysis [15,16]. Some studies have pointed out that reasonable risk allocation has a positive impact on project management performance. For example, Abednego and Ogunlana [17] clearly pointed out that reasonable risk allocation contributes to better project performance by improving the project governance level. However, more studies have pointed out that clients are more inclined to allocate risks in favor of themselves, such as forcing contractors to take excessive risks in the setting of contract terms [18]. Additionally, Tang et al. [19] pointed out that discussing pro-client risk allocation was more meaningful than discussing reasonable risk allocation, especially in construction projects. The reason is that in practice, many clients assign too many risks to contractors, and contractors' opportunistic behaviors could potentially arise from the transfer of clients' risk responsibility. Therefore, the research object of this study was defined as pro-client risk allocation, which refers to unreasonable risk allocation behaviors such as allocating excessive risks to contractors, including risks that clients should bear, or allocating all external risks, such as force majeure to contractors, and further revealing its influence on contractors' opportunistic behaviors.

### 2.2. Trust

As an object of study, trust has achieved fruitful research results in economics, management, organizational behavior, psychology, and many other fields. Trust itself contains many definitions, which are easy to change according to the environment [20]. Most scholars understand it in terms of trusting the trustee based on optimistic expectations, and the trustee is in a psychological state of being vulnerable to accidental damage [21]. Under the guidance of the trust relationship in projects, the hostile contradiction between clients and contractors will be eased, and opportunistic behavior will be reduced [22]. In the process of inter-organizational cooperation, trust promotes both parties in a contract to move toward equal cooperative relations, forming a working pattern of mutual



trust [23]. Wong and Cheung emphasized that trust can establish and consolidate good cooperation and the relationship between development and contracting parties [24]. Teo et al. proposed to reduce opportunistic behavior by enhancing trust in contracting and embedding joint relationships in project networks [25].

### 2.3. Control

Control refers to measures and systems used to ensure that the organization's behavior is consistent with the project objectives [26]. Inter-organizational control refers to the control between participating parties in the transaction process, focusing on standardizing and coordinating the behaviors of both parties to ensure the realization of expected goals [27]. Control in construction projects usually refers to contract-based control. By signing a contract, the client and the contractor set up a formal control mechanism to restrain the behavior of both parties in the project. The client can specify the standards of output or procedures and observe and confirm if the contractor breaches the contract through the control mechanism [28]. Asymmetric information and the characteristics of a non-repeated game make both parties tend to maximize their own interests in the process of cooperation, which easily leads to opportunistic behaviors and increases transaction risks. A good contract control mechanism can reduce the uncertainty of both parties' perception of the project, effectively regulating and coordinating their behaviors to ensure the realization of project objectives [29].

### 2.4. Opportunistic Behaviors

TCE defines opportunism as "the act of obtaining one's own interests by fraud" [30,31], which is specifically described as "the act of misleading or confusing others by distorting, obscuring, forging or concealing information, etc., so as to obtain unilateral interests at the expense of others' interests". Opportunistic behaviors exist in many fields, such as scientific and technological innovation, environmental protection, enterprise management, engineering construction, etc. Opportunistic behavior can lead to major accidents, resulting in loss of personnel and property, etc. [32,33]. Governing and avoiding opportunistic behavior is an important issue in practical and academic circles. In construction projects, opportunistic behaviors of contractors can lead to project disputes, which can increase the actual transaction costs of both clients and contractors in the implementation of projects, lead to delays in the construction period, reduce project quality, and have an impact on the cooperation between the two parties [34].

Luo [35] put forward two concepts of opportunism, namely the strong and the weak. Strong opportunism mainly refers to violating a specific contract or supplementary agreement provision, while weak opportunism refers to violating relationship norms, which are not stipulated in contracts but rooted in the consensus of all members of a relationship. Strong opportunistic behaviors are more likely to be observed or detected and remedied than weak opportunistic behaviors [4]. Contracts usually provide a way for both parties to resolve disputes. In contrast, there is no clear punishment or basis for violating relationship norms. In construction projects, both strong and weak opportunistic behaviors exist, which will have serious consequences to the cooperation of both parties involved in the transaction, increase the cost of the transaction, and weaken the cooperative relationship between the parties [36]. However, the influence mechanism of risk allocation in strong and weak opportunistic behaviors is different, and the control measures are not completely consistent [37]. Therefore, in construction projects, it is more meaningful to study opportunistic behaviors in two dimensions, strong and weak, than in a single dimension. The division of strong and weak opportunistic behaviors is also more consistent with engineering practices [38]. In this study, opportunistic behavior was divided into strong and weak dimensions and measured in combination with the actual situation of projects.

## 3. Conceptual Framework and Hypothesis Development

### 3.1. Risk Allocation and Opportunistic Behaviors

Zhang et al. [39] pointed out that reasonable risk allocation clauses can promote the fair perception of contractors and cooperative behaviors of both parties. The risk allocation process of contracts, to a certain extent, is also a process of balancing the interests of project participants and maximizing the interests of project arrangement. Meanwhile, Levitt et al. [40] found that reasonable risk allocation can save project costs and time to the ultimate conclusion and achieve a win-win situation through many empirical studies of construction projects. However, in practice, due to the clients' bargaining power in contract negotiations, clients will allocate most of the risks to contractors in contracts [39], which means a potential loss to the contractors. Contractors think that it is difficult to offset the economic losses caused by risks by the price compensation they get. Thus, to make up for the loss, they may adopt defensive strategies, which may lead to strong opportunistic behaviors such as not fully fulfilling their responsibilities and obligations [5,41]. Consequently, Hypothesis H1 was proposed:

**Hypothesis 1 (H1).** *Pro-client risk allocation increases strong opportunistic behavior.*

Allocating most risks to contractors will worsen the relationship between clients and contractors. Taking too many risks will lead to a sense of unfairness among contractors, which will lead to a breakdown of trust and mutual suspicion between the two parties, leading to less willingness to cooperate [42]. Contractors are unwilling to cooperate in the production period or even in conflicts with the existing cooperation relationship, which may lead to a lack of enthusiasm in their performance, negative work behavior, or the use of contract loopholes and unbalanced quotation by concealing of information and other weak opportunistic behaviors [43]. Thus, Hypothesis H2 was proposed:

**Hypothesis 2 (H2).** *Pro-client risk allocation increases weak opportunistic behavior.*

### 3.2. Moderating Effect of Trust

Relationship governance theory holds that a transaction is an interactive process that requires formal contracts to regulate transaction behaviors and a relationship governance mechanism to manage the relationship between transaction objects, and trust is an important concept of the relational governance mechanism [16]. Shi et al. [4] defined trust as the degree to which a trading party believes in the honesty and reliability of a partner. In construction projects, the higher the degree of trust between clients and contractors, the stronger the mutual dependence [44].

Contractors' trust in clients means that they believe in and rely on clients and are willing to accept the clients' behavior [45]. This trust indicates that contractors are willing to accept and understand the risks involved in contracts [46]. A high degree of trust gives contractors confidence that clients will perform their contractual duties [24], and this triggers contractors to act on their own initiative. At the same time, trust is an important factor affecting the willingness of both parties to continue cooperation [47]. In an environment with a high level of trust, contractors will not engage in strong opportunism behavior to destroy a contract for short-term benefit. This leads to Hypothesis H3a:

**Hypothesis 3a (H3a).** *Trust moderates the effect of pro-client risk allocation on the strong opportunistic behavior of contractors.*

With increased trust, both clients and contractors are more likely to exchange views more frequently and openly, disclose more accurate information, and have less information asymmetry [48,49]. Therefore, trust can promote cooperation between both parties of the transaction and weaken contractors' tendency toward opportunism [50]. Clients' confidence in contractors means that they believe contractors are capable of completing projects and are willing to cooperate in good faith [15]. Meanwhile, contractors

also have more confidence in clients' investment and payment ability [51]. In the long run, even if there are contract loopholes and unreasonable risk allocation, contractors are more willing to maintain common understanding and expectations in order to maintain long-term cooperative relations, rather than engage in weak opportunistic behaviors to exploit contract loopholes. Thus, Hypothesis H3b was proposed:

**Hypothesis 3b (H3b).** *Trust moderates the impact of pro-client risk allocation on the weak opportunism behavior of contractors.*

### 3.3. Moderating Effect of Control

According to project governance theory, management control can effectively mitigate transaction risks and promote a cooperative relationship between the two parties [15,31]. In construction projects, control refers to the management and supervision procedures that influence the behavior of contractors to achieve the established objectives. Studies of control mechanisms are usually classified into two categories: formal and informal control [15]. Formal control includes contractual provisions and formal organizational arrangements, which can be subdivided into process control and outcome control. Informal control, also known as social control, refers to reducing the differences in target preferences between trading parties by establishing common values and beliefs so as to promote self-control between the two parties. Informal control can be achieved through communication, meetings, and other activities. When clients adopt appropriate control measures, project performance can be significantly improved. When contractors assume excessive risks and generate an opportunistic tendency, effective control means controlling the occurrence of opportunistic behaviors [52].

The more complete the contract terms are, the more difficult it is for contractors to engage in opportunistic behaviors, especially strong opportunistic behaviors that directly violate the contract. Contract terms with good integrity have strong control over contractors, and penalty measures are often specified in detail in the contract. Contractors must consider the legal and economic consequences before engaging in strong opportunistic behaviors [53] so they can effectively restrain defense behaviors caused by risk allocation. Clients engage in process control, such as participating and intervening in contractors' daily activities, and strict control and review of contractors' daily work can also reduce the chance for contractors to adopt strong opportunism [54]. In view of this, Hypothesis H4a was proposed:

**Hypothesis 4a (H4a).** *Control moderates the effect of pro-client risk allocation on the strong opportunistic behavior of contractors.*

Contract control can also reduce the positive influence of risk allocation on weak opportunism behavior. As the risk taken by contractors increases, contractors are likely to directly violate contracts and are also more likely to take advantage of contract loopholes and violate the norms of the relationship that are not clearly defined in contracts [37]. When contract terms are set strictly, contractors can exploit fewer loopholes, and the opportunism behavior caused by risk allocation is weak. At the same time, contracts can also provide certain terms, such as how the parties communicate and share information, to avoid coordination failures and misunderstandings [55]. The parties will exchange information to promote common understanding and expectations [36], thereby reducing information asymmetry and weakening the link between risk allocation and weak opportunistic behavior [30]. In view of this, Hypothesis H4b was proposed:

**Hypothesis 4a (H4b).** *Control moderates the effect of pro-client risk allocation on the weak opportunistic behavior of contractors.*

To sum up, Figure 1 depicts the hypotheses of this study. Hypotheses 1 and 2 focus on the direct relationship between risk allocation and opportunistic behavior, and Hypotheses 3 and 4 look at the moderating effect of trust and control on risk allocation on opportunistic behavior.

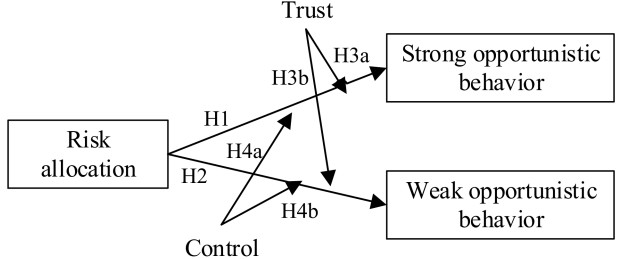

**Figure 1.** Hypothesized model.

## 4. Methodology

### 4.1. Data Collection

The questionnaire format was based on anonymous self-filled responses, and a 5-level Likert scale was used to measure risk allocation, opportunistic behavior, trust, and control. The target interviewees of this study were clients, contractors, and engineering consulting units involved in construction projects. Respondents were asked to answer questions about a specific project they were working on or had recently participated in. The questionnaire emphasized that there were no right or wrong answers and that all data would remain anonymous and confidential.

A total of 1000 questionnaires were distributed, and 360 were recovered. In order to ensure the validity of the samples, questionnaires completed within 5 min were eliminated (based on the estimated completion time). Finally, 342 valid samples were obtained in this study. In order to control the omitted variable bias, the questionnaire also collected basic information about the projects, including project type and duration and the contract price, as shown in Table 1. In addition, the results of ANOVA show that there were no significant differences in the answer data at the 95% confidence level. Therefore, different items did not have significant differences in the survey results.

**Table 1.** Characteristics of respondents and their projects.

| Characteristic | Range | Number | Percentage (%) |
|---|---|---|---|
| Project type | Building construction | 246 | 71.9 |
| | Public utility | 57 | 16.7 |
| | Port and waterway | 3 | 0.9 |
| | Others | 36 | 10.5 |
| Project construction period (years) | <2 | 114 | 33.3 |
| | 2–5 | 211 | 61.7 |
| | 6–10 | 15 | 4.4 |
| | >10 | 2 | 0.6 |
| Contract price (million RMB) | <30 | 62 | 18.1 |
| | 30–100 | 85 | 24.9 |
| | 100–1000 | 152 | 44.4 |
| | 1000–3000 | 21 | 6.1 |
| | >3000 | 22 | 6.4 |

### 4.2. Measurement

Variable measurement clauses used in this paper had high reliability and validity and had been confirmed in the relevant literature. Based on existing studies, this study combined the specific situation of construction practice in China to form an initial scale. After asking practitioners, experts,

and scholars in the industry for their suggestions on the scale, we further modified and improved it so as to determine the final scale.

### 4.2.1. Independent Variable: Risk Allocation

As for the specific measure of risk allocation, this study adopted four items proposed by Tang et al. [19] to measure risk allocation: design defects, the accuracy of information provided by the client, force majeure, and rising prices.

### 4.2.2. Dependent Variable: Opportunistic Behaviors

In this study, four items proposed by Luo [35] and three by Shi et al. [4] were respectively used to describe strong and weak opportunistic behaviors. Strong opportunistic behavior includes not acting according to the contract text or agreement, unilaterally suspending or terminating the performance of the contract, committing fraud when sharing important information required by the contract, and shirking responsibility that is not specified in the contract but is actually borne by the contractor. Weak opportunism includes incompletely disclosing certain information, providing work that fails to deliver on verbal promises, and exploiting loopholes in the contract for their own benefit.

### 4.2.3. Moderating Variables

#### Trust

Four items proposed by Cheung et al. [56] and Pinto, Slevin, and English [16] were used to measure trust: believing that the contractor is capable of performing his duties, trusting that the contractor will keep his promises during the project, trusting that the contractor will take his interests into account when making decisions, and understanding that good personal relationships can improve the working relationship between client and contractor.

#### Control

Four items proposed by Wang et al. [57] and Tang et al. [19] were used to measure control: the establishing of a clear working procedure, continuously supervising the implementation of the contractor's plan and construction progress, setting clear performance targets, and linking rewards and penalties to the achievement of performance goals.

### 4.2.4. Control Variables

In addition to the above variables, this study also considered other factors affecting opportunistic behavior as control variables: job position, working years, project type, project duration, and contract price [58–60].

### 4.3. Measurement Validation

In order to assess the internal consistency and reliability of the scales, Cronbach's alpha was tested with SPSS 25.0, and the results are as shown in Table 2. Cronbach's alpha > 0.7 and the composite reliability (CR) of the five variable measurement items in Table 2 are all higher than 0.7, indicating good internal consistency and reliability [61]. To test the validity of the scale, this study used Mplus v8.3 to conduct confirmatory factor analysis (CFA) by $\chi^2$/df = 2.34 < 3, RMSEA = 0.063 < 0.08, TLI = 0.939 > 0.9, CFI = 0.949 > 0.9, and SRMR = 0.041 < 0.08, and the model had a good fit [62]. Aggregate validity was used to measure the intensity represented by the item, expressed by average variance (AVE). The AVE values in this study are all higher than 0.5, and as shown in Table 3, the square root of AVE exceeds the correlation of other potential variables (off-diagonal elements), which means that the validity of this study is good [63].

**Table 2.** Measures of reliability and validity assessment.

| Construct and Measuring Items | SFL |
|---|---|
| RA ($\alpha$ = 0.876; AVE = 0.6401; CR = 0.8767) | |
| RA1. Contractors bear most of the risk caused by defective design. | 0.783 |
| RA2. Contractor bears most of the risk arising from accuracy of instructions or information. | 0.834 |
| RA3. Contractor bears most of the risk arising from unforeseeable physical conditions. | 0.802 |
| RA4. Contractor bears most of the risk arising from inflation of prices. | 0.780 |
| SOB ($\alpha$ = 0.856; AVE = 0.6027; CR = 0.8574) | |
| SOB1. Contractor has not acted in accordance with the text or agreement of the contract. | 0.692 |
| SOB2. Contractor unilaterally suspends or terminates the performance of the contract. | 0.823 |
| SOB3. Contractor commits fraud when sharing important information required by the contract. | 0.868 |
| SOB4. Contractor shirks the responsibility that is not specified in the contract but is actually his to assume. | 0.708 |
| WOB ($\alpha$ = 0.864; AVE = 0.6803; CR = 0.8645) | |
| WOB1. Contractor does not disclose certain information to protect his own interests. | 0.830 |
| WOB2. Contractor sometimes fails to perform work that is verbally promised. | 0.839 |
| WOB3. Contractor will try to exploit loopholes in contracts for his own gain. | 0.805 |
| T ($\alpha$ = 0.867; AVE = 0.6372; CR = 0.8734) | |
| T1. Client believes that contractor is competent enough to perform his duties. | 0.875 |
| T2. Client trusts that contractor will keep his promises in the course of the project. | 0.906 |
| T3. Client trusts contractor to take his interests into account when making decisions. | 0.747 |
| T4. Client understands that building good personal relationships can improve the working relationship between client and contractor. | 0.636 |
| C ($\alpha$ = 0.895; AVE = 0.6859; CR = 0.8972) | |
| C1. Client has established clear working procedures that must be strictly observed by contractor. | 0.806 |
| C2. Client shall continuously supervise implementation of contractor's plan and construction progress. | 0.822 |
| C3. Client has set clear performance objectives for contractor. | 0.873 |
| C4. Client will link contractor's rewards and penalties to performance targets achieved. | 0.810 |

SFL: standardized factor loading; $\alpha$: Cronbach's alpha; AVE: average variance extracted; CR: composite reliability; RA: risk allocation; SOB: strong opportunistic behavior; WOB: weak opportunistic behavior; T: trust; C: control.

**Table 3.** Means, standard deviations, and correlations.

| Variable | M | SD | 1 | 2 | 3 | 4 | 5 |
|---|---|---|---|---|---|---|---|
| Strong opportunistic behavior | 2.62 | 0.85 | **0.776** | | | | |
| Weak opportunistic behavior | 3.08 | 0.96 | 0.446 ** | **0.825** | | | |
| Risk allocation | 3.13 | 0.85 | 0.170 * | 0.135 * | **0.800** | | |
| Trust | 3.74 | 0.74 | −0.169 ** | −0.089 | 0.105 | **0.798** | |
| Control | 3.68 | 0.82 | −0.138 * | −0.131 * | 0.101 | 0.614 *** | **0.828** |

Notes: Boldface indicates values that are greater than off-diagonal correlations. * $p < 0.05$; ** $p < 0.01$; *** $p < 0.001$.

### 4.4. Path Inspection

Hierarchical regression analyses were carried out in SPSS 25.0 to test the hypotheses. The independent and moderating variables were restored by the mean centering technique to reduce multicollinearity when testing the moderating effects. The variance inflation factor (VIF) values of all independent and control variables range from 1.011 to 1.245, all less than 10, indicating that no multicollinearity disturbed the results [64].

### 4.4.1. Main Effect

The effect of risk allocation on strong opportunistic behaviors is shown in Table 4, panel A. Model A2 describes the distribution of the influence of risk on strong opportunism behavior; as can be seen from the model A2, risk allocation has a significant positive effect on strong opportunistic behavior ($\beta$ = 0.172, $p < 0.01$), supporting Hypothesis H1.

**Table 4.** Results of regression analysis.

| Panel A | Strong Opportunistic Behaviors (SOBs) | | | | | | | |
| --- | --- | --- | --- | --- | --- | --- | --- | --- |
| | Model A1 | | Model A2 | | Model A3 | | Model A4 | |
| | β | VIF | β | VIF | β | VIF | β | VIF |
| Control variables | | | | | | | | |
| Project type | 0.084 | 1.022 | 0.076 | 1.024 | 0.064 | 1.028 | 0.055 | 1.032 |
| Project construction period | 0.038 | 1.234 | 0.041 | 1.234 | 0.03 | 1.242 | 0.033 | 1.245 |
| Contract price | 0.025 | 1.209 | 0.029 | 1.210 | 0.021 | 1.213 | 0.015 | 1.213 |
| Independent variables | | | | | | | | |
| Risk allocation(RA) | | | 0.172 *** | 1.011 | 0.22 *** | 1.044 | 0.198 *** | 1.025 |
| Moderating variables | | | | | | | | |
| Trust(T) | | | | | −0.202 *** | 1.016 | | |
| Control(CO) | | | | | | | −0.152 *** | 1.024 |
| Interaction | | | | | | | | |
| RA × T | | | | | −0.186 *** | 1.042 | | |
| RA × CO | | | | | | | −0.184 *** | 1.025 |
| F | | 0.761 | | 2.356 ** | | 5.163 *** | | 4.509 *** |
| ΔF | | 0.761 | | 10.223 ** | | 12.423 *** | | 12.183 *** |
| R2 | | 0.011 | | 0.04 | | 0.11 | | 0.98 |
| ΔR2 | | 0.011 | | 0.029 | | 0.033 | | 0.033 |
| Panel B | Weak Opportunistic Behaviors (WOBs) | | | | | | | |
| | Model B1 | | Model B2 | | Model B3 | | Model B4 | |
| | β | VIF | β | VIF | β | VIF | β | VIF |
| Control variables | | | | | | | | |
| Project type | 0.113 | 1.022 | 0.07 | 1.024 | 0.062 | 1.028 | 0.06 | 1.032 |
| Project construction period | 0.076 | 1.234 | 0.087 | 1.234 | 0.079 | 1.242 | 0.089 | 1.245 |
| Contract price | 0.075 | 1.209 | 0.07 | 1.21 | 0.064 | 1.213 | 0.062 | 1.213 |
| Independent variables | | | | | | | | |
| Risk allocation (RA) | | | 0.145 *** | 1.011 | 0.174 *** | 1.044 | 0.162 *** | 1.025 |
| Moderating variables | | | | | | | | |
| Trust (T) | | | | | −0.118 ** | 1.016 | | |
| Control (CO) | | | | | | | −0.139 *** | 1.024 |
| Interaction | | | | | | | | |
| RA × T | | | | | −0.118 ** | 1.042 | | |
| RA × CO | | | | | | | −0.057 | 1.025 |
| F | | 4.51 *** | | 4.337 *** | | 4.510 *** | | 4.576 *** |
| ΔF | | 4.51 *** | | 7.459 *** | | 4.947 ** | | 0.128 |
| R2 | | 0.63 | | 0.072 | | 0.098 | | 0.120 |
| ΔR2 | | 0.063 | | 0.021 | | 0.013 | | 0.000 |

Notes: * $p < 0.10$; ** $p < 0.05$; *** $p < 0.01$. β is standardized regression coefficient. VIF: variance inflation factor.

The effect of risk allocation on weak opportunism behavior is shown in Table 4, panel B. Model B2 describes risk allocation for the weak effect of opportunistic behavior; as can be seen from the model B2, risk allocation has a significant effect on weak opportunism behavior ($\beta = 0.21$, $p < 0.01$), supporting Hypothesis H2.

### 4.4.2. Moderating Effect

To test the moderating effect of trust and control, the interaction terms were added to model A2. Model A3 describes the moderating effect of trust on the strong opportunistic behavior of risk allocation. The result shows that the interaction effect of trust with risk allocation (RA × T) is negative ($\beta = -0.186$, $p < 0.01$). According to the interaction between trust and risk allocation, a simple slope analysis was drawn, as shown in Figure 2A. It indicates that trust has a negative regulatory effect on strong opportunistic behavior, supporting Hypothesis H3a.

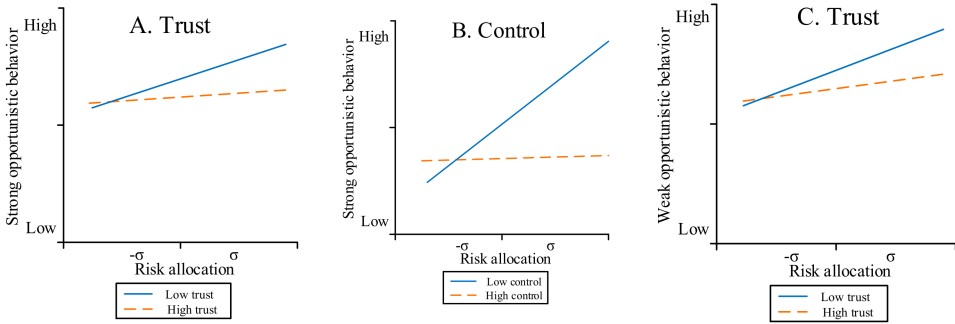

**Figure 2.** Graphical representation of moderation effects. (**A**) Moderation effect of trust between risk allocation and strong opportunistic behavior (**B**) Moderation effect of control between risk allocation and strong opportunistic behavior (**C**) Moderation effect of trust between risk allocation and weak opportunistic behavior.

Model A4 describes the moderating effect of control on the strong opportunistic behavior of risk allocation. The results show that the interaction of control and risk allocation (RA × CO) is negative ($\beta = -0.184$, $p < 0.01$). In order to further illustrate the moderating effect, we conducted a simple slope analysis of the interaction of control and risk allocation, as shown in Figure 2B. It indicates that control of strong opportunism behavior has a negative regulatory role, supporting Hypothesis H4a.

In order to test the moderating effect of trust and control on weak opportunistic behaviors, interactive items were added based on model B2. Model B3 describes trust for the regulation of the weak effect of opportunism risk allocation function. The results show that the interaction of trust and risk allocation (RA × T) is negative ($\beta = -0.118$, $p < 0.01$). According to the interaction of trust and risk allocation, we conducted a simple slope analysis, as shown in Figure 2C. It suggests that trust in weak regulation of opportunistic behavior has a negative effect, supporting Hypothesis H3b. Model B4 describes the control of the regulation of the weak effect of opportunism risk allocation function. The results show that the interaction of control and risk allocation (RA × CO) was not significant ($\beta = -0.057$, $p > 0.1$), which does not support Hypothesis H4b.

The summary of the hypothesis-testing results is shown in Table 5.

**Table 5.** Hypothesis-testing results.

| Hypothesis | Path of Hypothesis | $\beta$ | $p$ | Correlation | Results |
|:---:|:---:|:---:|:---:|:---:|:---:|
| H1 | Risk allocation→Strong opportunistic behaviors | 0.172 | 0.002 | positive | S |
| H2 | Risk allocation→Weak opportunistic behaviors | 0.145 | 0.007 | positive | S |
| H3a | Risk allocation × Trust→Strong opportunistic behaviors | −0.186 | 0.000 | negative | S |
| H3b | Risk allocation × Trust→Weak opportunistic behaviors | −0.118 | 0.027 | negative | S |
| H4a | Risk allocation × Control→Strong opportunistic behaviors | −0.184 | 0.001 | negative | S |
| H4b | Risk allocation × Control→Weak opportunistic behaviors | −0.057 | 0.285 | uncorrelated | NS |

S: supported; NS: not supported.

## 5. Result and Discussion

Based on previous studies, this study, from the perspective of project governance theory, constructed a model with risk allocation as the independent variable, opportunistic behavior as the dependent variable, and trust and control as the moderating variable. It clarifies how risk allocation affects the opportunistic behavior of contractors and confirms the moderating effect of trust and control.

### 5.1. Impact of Pro-Client Risk Allocation on Contractors' Opportunistic Behavior

Hypothesis H1 was confirmed, indicating that pro-client risk allocation has a positive effect on contractors' opportunistic behavior, which is consistent with research results in the United Kingdom, USA, Australia, Hong Kong, and other countries and regions [9,10,65,66]. These studies showed that

clients use exemption clauses to transfer a large majority of risks to contractors and avoid their own responsibilities. Such pro-client risk allocation will lead contractors to engage in opportunistic behavior during bidding and executing. The more risks assigned to contractors, the more significant their opportunistic behavior [67–69]. However, existing studies on the opportunistic effect of unreasonable risk allocation only analyzed the impact of risk allocation on opportunistic behavior, but did not analyze the characteristics of such behavior, ignoring its further classification. This study divided opportunistic behavior into two dimensions, strong and weak, and analyzed two types of opportunistic effects of pro-client risk allocation, which adds more details and deepens the existing research on the topic.

### 5.2. Moderating Role of Trust between Pro-Client Risk Allocation and Contractors' Opportunistic Behavior

The results of this study show that trust has a significant moderating effect on both strong and weak opportunistic behaviors. The results are consistent with those of Anvuur et al. [70,71]. These studies showed that in an environment where there is a high level of trust, contractors are willing to exchange information publicly and take the initiative to disclose the information to clients, such as pointing out loopholes in contracts. This study points out that clients should show trust in contractors' actions, and contractors should show due diligence and cooperation in return, so as to minimize the strong opportunistic behaviors that destroy contracts. At the same time, trust between clients and contractors can effectively reduce the occurrence of weak opportunistic behaviors in the process of cooperation, such as the contractor being lazy, evading, or not fully cooperating. During the questionnaire survey, some contractor respondents pointed out that if clients have full trust in contractors, then contractors will perceive the clients' trust and goodwill. Even if unreasonable risk allocation clauses are agreed to in contracts, contractors will fulfill their contractual obligations, avoid strong or weak opportunistic behavior, repay the trust of clients, and ensure the success of the project, with a view to long-term friendly cooperation between the two parties.

### 5.3. Moderating Role of Control Between Pro-Client Risk Allocation and Contractors' Opportunistic Behavior

Hypothesis H4a was confirmed, indicating that control has a negative moderating effect on strong opportunistic behavior, echoing the research of Shi et al. [4], which pointed out that a contract, as a formal control mechanism, specifies the rights and obligations of both parties in detail and increases the cost of opportunistic behavior on the part of contractors. The results of this study also show that this clear and legally effective control mechanism can effectively restrain contractors' obvious strong opportunistic behavior. However, assuming that H4b is not verified, this may be because weak opportunistic behaviors are relatively hidden and difficult to detect, and clear control measures have no obvious governance effect on this kind of opportunistic behavior. In the research process of this study, some client and contractor respondents reported that agreed-upon control clauses in contracts will obviously increase the cost of contractors' breach of contracts, which will make them afraid to engage in strong opportunistic behavior, but such control clauses will not have as much of an effect on restraining hidden weak opportunistic behavior.

## 6. Conclusions and Implications

### 6.1. Conclusions

This study revealed the impact of risk allocation on opportunistic behavior and explored the moderating effects of trust and control. The findings of 342 samples suggest that excessive risk-taking by contractors reinforces opportunistic behavior. In addition, this study proved that trust can reduce the positive impact of risk allocation on strong and weak opportunistic behaviors, but control can reduce the positive impact of risk allocation only on strong opportunistic behaviors and has no significant regulating effect on weak opportunistic behaviors. Therefore, when contractors

bear most of the risks, they should be given full confidence, and appropriate control measures should be taken.

## 6.2. Theoretical Implications

First, this study supplements the risk allocation literature by examining unreasonable risk allocation. Previous studies focused on how to restrain opportunism through reasonable risk allocation. There is a lack of research on how to dilute the role of opportunism when the risk is unreasonable and fixed. This study introduced trust and control as moderating variables to examine their diluting effect on the relationship between risk allocation and opportunism. The results point out that clients should show full trust in contractors, and in return contractors should be diligent and cooperative, thus effectively reducing their strong and weak opportunistic behavior. As a formal control mechanism, a contract stipulates the rights and obligations of both parties in detail, increases the cost of contractors' opportunistic behaviors, and can effectively restrain obvious strong opportunistic behaviors, while weak opportunistic behaviors are hidden and difficult to detect. Clear control measures have little effect on this kind of behavior.

Second, this paper expands the research of contractors' opportunist behavior. Existing studies on the opportunist effect of relationship risk allocation mostly analyzed the impact of risk allocation on opportunist behavior, without analyzing the characteristics of such behavior [39,72], ignoring their further classification. Based on project governance theory, when clients assign too much risk to contractors, contractors engage in opportunistic behavior, and this study divided opportunistic behavior into two dimensions based on strength and extended project governance theory analysis, deepening the existing research on the topic.

## 6.3. Managerial Implications

This study provides important knowledge for managers involved in construction projects. First of all, clients should consider the potential tendency for contractors to be opportunistic when making decisions about risk allocation. Clients can realize from this study that contractors may engage in strong or weak opportunistic behavior. In addition to violating formal agreements, contractors can also obtain benefits by violating oral agreements and taking advantage of contract loopholes. In order to choose a targeted governance mechanism for opportunistic behaviors, clients must clearly identify the type of opportunism. In construction projects, clients should be alert to both strong and weak opportunistic behaviors. According to the results of the study, trust can dilute the effect of unreasonable risk allocation on opportunism. Clients can reward contractors with more project control rights and corresponding profit margins. In practice, the engineering procurement construction (EPC) project reflects this point [73]. In an EPC project, the initial contractor bears too many risks, and his tendency to be opportunistic can be alleviated by having trust in the process of contract execution. In addition, the control function of the contract can only alleviate the effect of risk allocation on strong opportunism. Therefore, in reality, it is observed that clients control contractors' performance through penalty clauses in contracts, but this cannot control contractors' perfunctory behavior because this is weak opportunism and cannot be written into contracts. Therefore, the penalty clauses in contracts are only suitable for those projects with clear results [74].

Second, in order to reduce the opportunistic behaviors of contractors, clients should combine trust and control. The results show that when clients assign most of the risks to contractors, their opportunistic behaviors can be reduced through the safeguards of trust and control. Therefore, in construction projects, in order to restrain strong and weak opportunistic behaviors, clients should use the values and concepts of contractors to establish a common vision for both parties so as to improve contractors' motivation and enthusiasm and work toward common goals. If contractors intentionally breach the terms and do not act in accordance with contracts, clients should take effective control measures such as clarifying the detailed rules of the contracts and listing as much as possible the penalties for breach of contracts so as to enhance the restraint on contractors. In analyzing the

positive impact of risk allocation on strong opportunism, there was no significant difference between trust and control regulation, but trust has a significant regulatory role in weak opportunistic behavior, therefore, clients should implement strict control measures and at the same time should strengthen the trust of contractors to promote their cooperation and reduce their opportunism tendency so as to promote project success.

*6.4. Limitations and Future Research*

This study has some limitations and needs further research. First, all of the respondents and project samples collected in this study are from China. Chinese culture may influence clients' perception of contractors' opportunistic behaviors and how to take measures. In the future, the influence of control and trust on risk allocation and opportunistic behavior in the Chinese context can be studied. Second, this paper examines the initial risk allocation of a project and does not consider risk reallocation in the process of working on a project, which can be studied later. Third, this paper studies the impact of trust and control on the opportunistic effect of risk allocation but not specific ways of displaying trust and control in depth. Future studies can elaborate on the measures of trust and control and explore the effects of different measures on risk allocation and opportunistic behavior. Finally, this study did not consider the effect of opportunistic behavior in other industries, thus future research can further explore this.

**Author Contributions:** The authors confirm contribution to the paper as follows: conceptualization, Y.Y., Q.L., and H.Y.; formal analysis, Q.L. and W.X.; investigation, W.X.; methodology, Q.L. and H.Y.; project administration, Y.Y.; resources, Y.Y.; software, W.X. and H.Y.; validation, Q.L. and W.X.; writing—original draft, Q.L. and W.X.; writing—review and editing, Y.Y. and Q.L. All authors have read and agreed to the published version of the manuscript.

**Funding:** This research was sponsored by the National Natural Science Foundation of China (Grant No. 71472135).

**Conflicts of Interest:** The authors declare no conflict of interest.

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
