# Peer review of "Impacts of Risk Allocation on Contractors’ Opportunistic Behavior: The Moderating Effect of Trust and Control"

_sustainability, doi:10.3390/su12229604_

Round 1
Reviewer 1 Report
The paper deals with the item of opportunistic behaviour in construction projects and falls within the broad scope of Sustainability. I appreciate the effort of the authors in investigating this conflictual and interesting topic. In fact, even if not completely new, the topic can be considered of interest, provided that authors improve the overall quality of their manuscript. In the following, I enclose some recommendations to raise the interest of the reader and improve style, structure and content.
- Introduction is clearly developed but the main objective of their study should be better clarified including the novelty in comparison with several existing studies on the same subject with similar approaches.
- Literature review should be distinguished from conceptual framework. Authors should try to expand state-of-the-art authors also to Countries different from China, including also some numerical figures and actual indicators for qualifying behaviour in the given industrial sectors and in other ones.
- The section Methodology (or Materials and methods) should include the research framework and all details regarding data, statistical elaboration, validation etc. A schematic block flow-diagram including Fig. 1 can help in describing the theoretical approach upon which this paper is based. Moreover, in sections 4. Results and 5. Discussion, there are several paragraphs that should be better clarified and included under the section Methodology.
- Results should be presented more clearly, mainly in commenting table 4, while as reported all details regarding the model, software tec. should be moved into the previous section.
- Discussion should include quantitative comparison with studies from different Countries and adequate reference, so as to contextualize the contribution of this paper. Also English style should be improved e.g. rewrite sentence line 317-320. A possible point of interest in relation to line 379-382, is the effect on safety of breaching terms of contract and the effect of opportunistic behaviour on incident statistics and safety performances, or in some case take to major accidents.
Also a comparison with similar problems in other sectors may be worthwile also to increase the possible interest for this study to other scientific domains. For example Bhopal major accident in the process sector is in some way connected to management of change and uncorrect behaviour of contractors during maintenance and upgrade of safety systems and authors can add ref to: http://dx.doi.org/10.1016/j.psep.2015.06.009 From the other side, opportunistic behaviour and the adoption of shortcomings, or non proper adoption of safety rules can result in higher occupational injury frequency for contractors (authors may add ref. to https://doi.org/10.1016/j.psep.2020.08.006
Managerial implications should move from qualitative to quantitative. Authors should give some indications regarding what they claim in lines 407-409 as future direction of their research, at least by proper referencing to existing studies on reward or penalty approach.
- Conclusions should be structured as a single paragraphs highlighting better the limitations and the several simplifying hypotheses upon which the study is conceived.
After final revision of the manuscript, English style should be checked by an English mother tongue ,also to delete minor typos.
Author Response
Thank you for giving us the opportunity to revise and resubmit our manuscript. We wish to thank the editors and reviewers for their time in effort in reviewing our manuscript. We have marked the changes using the "Track Changes" function in Microsoft Word. We hope the changes listed have made the manuscript suitable for publication and we look forward to your response.
Please see the attachment.

Reviewer 2 Report
Dear Authors,
Thank you for the interesting research. Several comments regarding the presented model please find bellow:
- The title of the manuscript is to long. It can be more reflected presented behavior analysis;
- The factors or risk allocation of risks in the manuscript can be entitle by appraisal more detail;
- Newly literature source from 2019-2020 year must be presented in the literature review and reference list;
- The selected focus group must be entitle for the behaviour analysis;
- The methodical part with a calculation processes can be described;
- Practical case study in construction project can be presented in the manuscript as an part of the manuscript.
Reviewer
Author Response
Thank you for giving us the opportunity to revise and resubmit our manuscript. We wish to thank the editors and reviewers for their time in effort in reviewing our manuscript. We have marked the changes using the "Track Changes" function in Microsoft Word. We hope the changes listed have made the manuscript suitable for publication and we look forward to your response.
Response to Reviewer 2 Comments
Point 1: The title of the manuscript is to long. It can be more reflected presented behavior analysis.
Response 1: Thank you for the suggestion! It pointed out the problem of the title. We highly recognized the opinion and realized that the title of the previous version was not fit for the research objective of this study very well. After in-depth discussion, we revised the title of this article as follows: “Impacts of Risk Allocation on contractors’ Opportunistic behaviour: the morderating effect of Trust and Control”. (see lines 2-4)
Point 2: The factors or risk allocation of risks in the manuscript can be entitle by appraisal more detail.
Response 2: Thank you for the suggestion! We are grateful for paying attention to this problem, but the methodology adopted in this study is factor analysis, and its essence is to be entitle each measurement item. The risk allocation of risks in this study was entitled in table 2, RA1-RA4. (see line 294)
Point 3: Newly literature source from 2019-2020 year must be presented in the literature review and reference list
Response 3: Thank you for the suggestion! In the process of revision, 23 new references were added. Among all the 75 references in this paper, there are 8 newly literature sources from 2019 to 2021, including 3 in 2019 (Reference No.22, 29, 58), 2 in 2020 (Reference No.13, 19) and 3 in 2021 (Reference No.7, 9, 33). The references in 2021 have been downloaded from the journal official website and will be published in January 2021. The added newly literature sources from 2019-2021 include:
Chen Yilin;Yin Yilin. A study of the improvement of project management performance by flexible contract based on trust. Science Research Management 2019, 40, (03), 197-208.
DU Ya-ling; KE Dan; ZHANG Kai-hong. A scenario simulation experimental study on the impact of control on trust during the implementation of PPP contract: The moderating effects of role perception. Journal of Industrial Engineering and Engineering Management 2019, 33, (04), 97-103.
Wang, D.; Fang, S.; Fu, H., Impact of Control and Trust on Megaproject Success: The Mediating Role of Social Exchange Norms. Advances in Civil Engineering 2019, (PT.1), 1-12.
Castelblanco, G.; Guevara, J.; Mesa, H.; Flores, D. Risk Allocation in Unsolicited and Solicited Road Public-Private Partnerships: Sustainability and Management Implications. Sustainability 2020, 12, (11), 10.3390/su12114478.
Tang, Y.; Chen, Y.; Hua, Y.; Fu, Y., Impacts of risk allocation on conflict negotiation costs in construction projects: Does managerial control matter? International Journal of Project Management 2020, 38, 188-199.
Zhang Shuhua, Li Jinghuan, Li Yu, et al. Revenue Risk Allocation Mechanism in Public-Private Partnership Projects: Swing Option Approach. Journal of Construction Engineering and Management 2021, 147(1), 10.1061/(ASCE)CO.1943-7862.0001952.
Su Guiliang, Hastak Makarand, Deng Xiaomei, et al. Risk Sharing Strategies for IPD Projects: Interactional Analysis of Participants’ Decision-Making. Journal of Management in Engineering 2021, 37(1), 10.1061/(ASCE)ME.1943-5479.0000853.
Rongchen Zhu,Xiaofeng Hu,Jiaqi Hou,Xin Li. Application of machine learning techniques for predicting the consequences of construction accidents in China. Process Safety and Environmental Protection 2021,145, 293-302. https://doi.org/10.1016/j.psep.2020.08.006.
Point 4: The selected focus group must be entitle for the behaviour analysis.
Response 4: Thank you for the suggestion! We didn't mention any concept of "focus group" in this paper, and failed to understand and response this question accurately. Can you further explain and clarify this problem? Thanks very much!
Point 5: The methodical part with a calculation processes can be described.
Response 5: Thank you for the suggestion! In this study, SPSS25.0 and Mplus v8.3 were used to analyze 342 valid questionnaires. After we import the original data, the software automatically completed the calculation process. In the latest articles of Sustainability using structural equation modeling (SEM), such as Https://doi.org/10.3390/su12219185, Https://doi.org/10.3390/su12219088, Https://doi.org/10.3390/su12219019, Https://doi.org/10.3390/su12218879, etc., generally only report the calculation results by software, but do not describe the calculation process. If needing, we can provided all original datas recovered from this study, also can show the whole process of calculation and analysis of the application software in detail by recording a video.
Point 6: Practical case study in construction project can be presented in the manuscript as an part of the manuscript.
Response 6: Thank you for the suggestion! The research method of this study is empirical verification based on questionnaire, instead of case study method, and no in-depth investigation and analysis of construction project cases have been carried out. However, in the process of issuing and collecting questionnaires, some of the respondents shared their work experience with us. According to this comments, we added some feedback information of the respondents about practical cases in the discussion part of the revised version (see lines 367-371, lines 381-384).
Thank you for the valuable comments and suggestions. We wish the current version will meet with approval and greatly appreciate your effort in reviewing our manuscript.

Round 2
Reviewer 1 Report
Authors improved significantly the structure and content of their manuscript, addressing the major part of reviewers' comments.
A minor revision is requested regarding to check the correct format of the manuscript.
In Table 4 some numerical results are wrongly splitted into two lines.
There is no reason for duplicating Fig. 2C.
Author Response
We wish to thank the editors and reviewers for their time in effort in reviewing our manuscript. We have marked the changes in red in the tracked changes version. We hope the changes listed have made the manuscript suitable for publication and we look forward to your response.
Point 1: In Table 4 some numerical results are wrongly splitted into two lines.
Response 1: Thank you for pointing this out. Because the previous columns in Table 4 were not wide enough, some numerical results were splitted into two lines. After modification, all the numerical results in table 4 are in one line. (see line 311)
Point 2: There is no reason for duplicating Fig. 2C.
Response 2: Thank you for the suggestion. We have deleted the duplicating Fig.2C. Additionally, we resized figure 2a, 2b and 2C so that the three figures were placed on one line. (see line 319)
Thank you for the valuable comments and suggestions. In addition, we have carefully checked the grammar, spelling and punctuation etc. throughout the paper once again. We wish the current version will meet with approval and greatly appreciate your effort in reviewing our manuscript.
